# Juglone Inhibits *Listeria monocytogenes* ATCC 19115 by Targeting Cell Membrane and Protein

**DOI:** 10.3390/foods11172558

**Published:** 2022-08-24

**Authors:** Yingying Cai, Guangming Zou, Meihua Xi, Yujie Hou, Heyu Shen, Jingfang Ao, Mei Li, Jun Wang, Anwei Luo

**Affiliations:** College of Food Science and Engineering, Northwest A&F University, Yangling 712100, China

**Keywords:** juglone, *Listeria monocytogenes*, antibacterial activity, membrane damage

## Abstract

Foodborne disease caused by *Listeria monocytogenes* is a major global food safety problem. A potential solution is the antimicrobial development of the highly bioactive natural product juglone, yet few studies exist on its antibacterial mechanism against *L. monocytogenes*. Thus, we aimed to elucidate the antibacterial mechanism of action of juglone against *L. monocytogenes* by determining the resultant cell morphology, membrane permeability, membrane integrity, and proteome changes. The minimum inhibitory concentration of juglone against *L. monocytogenes* was 50 μg/mL, and *L. monocytogenes* treated with juglone had longer lag phases compared to controls. Juglone induced *L. monocytogenes* cell dysfunction, leakage of potassium ions, and membrane potential hyperpolarization. Confocal laser scanning microscopy and field-emission-gun scanning electron microscope assays revealed clear membrane damage due to juglone treatment. Fourier transform infrared analyses showed that *L. monocytogenes* responded to juglone by some conformational and compositional changes in the molecular makeup of the cell membrane. Sodium dodecyl sulfate-polyacrylamide gel electrophoresis analysis showed that juglone either destroyed proteins or inhibited proteins synthesis in *L. monocytogenes*. Therefore, our findings established juglone as a natural antibacterial agent with potential to control foodborne *L. monocytogenes* infections.

## 1. Introduction

*Listeria monocytogenes* is a Gram-positive, non-spore-forming bacterium that can acclimatize to various environments, including extreme conditions such as low pH (<4.4), water activity (<0.94), and temperature (4/45 °C) [1]. Thus, *L. monocytogenes* generally contaminate meat, vegetables, fish, processed and ready-to-eat food, and dairy products [2,3]. As a human and animal pathogen, *L. monocytogenes* is especially harmful to pregnant women, newborns, the elderly, and people with low immune function, causing life-threatening infections such as sepsis, encephalitis, and meningitis [4]. At these risks, the pathogen mortality rate can be as high as 20–30% [5]. Despite the current use of beta lactams alone or coupled with aminoglycosides to treat *L. monocytogenes* infections [6,7]. These therapies are becoming increasingly ineffective due to antimicrobial resistance [8,9], necessitating the development of new and effective antimicrobial agents against *L. monocytogenes*. Thus, we investigated the antimicrobial effects of juglone against *L. monocytogenes* in this study.

Many natural bioactive compounds are potential antibacterials with use that does not engender antimicrobial resistance [10]. One of these is juglone, also called 5-Hydroxy-1,4-naphthoquinone (C_10_H_6_O_3_), which is yielded by the walnut green husk. Not only is it lowly toxic and affordable [11,12], it also has good anticancer, antifungal, and antibacterial activities. Juglone is regarded as important drug candidate for the treatment of cancer, which had been proven to inhibit prostate cancer [13], cervical cancer [11], breast cancer [14], etc. Moreover, the antifungal activity of juglone was founded against *Candida albicans*, *Penicillium* spp., *Aspergilluus* spp., *Hansenula* spp., and *Saccharomyces carlsbergensis*. [15,16]. Gumus et al. (2020) reported that juglone is effective in clearing biofilm infections caused by *Candida albicans*. Additionally, there are some reports on juglone inhibiting bacteria [15]. Wang et al. (2016) analyzed differential proteins of *Escherichia coli* treated with juglone based on label-free quantitative proteomics, and the study suggested that juglone effectively restricted energy generation and blocked RNA formation and ribosome assembly [17]. Relatedly, juglone lysed *Staphylococcus aureus* by inhibiting pathways involved in protein synthesis, the tricarboxylic acid cycle, and nucleic acid synthesis [18].

Despite this proven antibacterial effect, there exist few studies on the inhibition of pathogens by juglone, and little is known about its antibacterial effect on *L. monocytogenes*. Thus, in this study, the inhibition zones, the minimum inhibitory concentrations (MIC), and growth-curve analyses of juglone against *L. monocytogenes* were determined to assess its antimicrobial activity. Then, membrane permeability was also analyzed by the leakage of potassium ions and membrane potential changes. The membrane integrity was further evaluated by confocal laser scanning microscopy (CLSM), field-emission-gun scanning electron microscope (FEG-SEM) assays, and Fourier transform infrared (FT-IR) analyses. Finally, we explored the effects of juglone on the pathogenicity of *L. monocytogenes* by analyzing changes in cellular proteins by sodium dodecyl sulfate-polyacrylamide gel electrophoresis (SDS-PAGE). Our findings established that juglone is a potential antibacterial agent for controlling foodborne *L. monocytogenes* infections.

## 2. Materials and Methods

### 2.1. Reagents

Juglone (≥98%, CAS: 481-39-0) extracted from walnut green husk was purchased from Shanghai Yuanye Bio-Technology Co., Ltd. (Shanghai, China). Stock solutions of juglone were prepared in 2% (*v*/*v*) ethanol, as previously described [19]. All other chemicals used were analytical grade.

### 2.2. Bacterial Strains and Growth Conditions

*L. monocytogenes* ATCC 19,115 was purchased from the American Type Culture Collection (Manassas, VA, USA) and cryopreserved at −80 °C [20]. Bacteria were cultured on trypticase soy agar (TSA; Land Bridge, Beijing, China) at 37 °C for 30 h. A single colony was incubated with shaking at 130 rpm for 18–24 h at 37 °C in 30 mL of sterile tryptic soy broth (TSB; Land Bridge, Beijing, China). The cell pellet was obtained by centrifugation (8000× *g*, 10 min, 4 °C) and rinsed twice with sterile TSB. Then, the pellet was resuspended in TSB, diluted to an OD_600nm_ = 0.5 (~10^8^ CFU/mL), and used in subsequent assays.

### 2.3. Antimicrobial Susceptibility Testing

The inhibition effect of juglone against *L. monocytogenes* was tested by Oxford cup method [21], with some modifications. Briefly, *L. monocytogenes* (200 μL) cells were plated on TSA. An Oxford cup (Φ7.8 × 6 × 10 mm) loaded with 200 μL juglone of 250, 500, 1000, 2000, and 4000 μg/mL was placed on the surface of the TSA plate. Absolute ethanol and ampicillin Na (50 μg/mL) were used as negative and positive controls, respectively. The plates were incubated at 37 °C for 24 h. The diameters of the inhibition zones were measured by a cursor caliper.

### 2.4. Determination of Minimum Inhibitory Concentration (MIC)

The minimal inhibitory concentration of juglone against *L. monocytogenes* was determined using the broth dilution method based on the Clinical and Laboratory Standards Institute guidelines [22]. Briefly, a 100-fold dilution of *L. monocytogenes* in TSB was made, and seven test tubes were inoculated with 5 mL of it. To these, juglone solution was added to create a dilution series with final juglone concentrations of 400, 200, 100, 50, 25, 12.5, and 6.25 μg/mL. The blank control, negative controls and positive controls were TSB-medium-only, juglone-free bacteria cultures containing 2% ethanol (*v*/*v*), and juglone-free bacteria cultures with ampicillin Na (50 μg/mL). All test tubes were mixed and incubated at 37 °C for 24 h. The MIC was defined as the lowest concentration of juglone that inhibited visually observable bacterial growth.

### 2.5. Growth Curves

Growth curves were constructed, as described by Shi et al. [23]. Briefly, *L. monocytogenes* was cultured overnight at 37 °C. The cells were rinsed with sterile TSB then diluted to an OD_600nm_ = 0.5 (~108 CFU/mL). A 100-fold dilution of this was used to make a microdilution series as follows. Bacteria cultures (125 μL) were mixed with 125 μL juglone solutions of various concentrations in a 100-well microtiter plate. This created a dilution series with final juglone concentrations of 200, 100, 50, 25, 12.5, 6.25, 3.13, 1.56, 0.78, and 0.39 μg/mL. Juglone-free cultures containing 2% ethanol (*v*/*v*) were used as controls. The samples were incubated for 24 h at 37 °C, with OD_600nm_ measurements automatically taken after every hour using a Bioscreen C system (Labsystems, Helsinki, Finland).

### 2.6. Potassium Ion Leakage

To determine whether juglone resulted in the release of potassium ions from bacterial cells and into the media, a previously described method was used [24]. Briefly, *L. monocytogenes* was cultured overnight at 37 °C, after which cells were rinsed and resuspended at a concentration of OD_600nm_ = 0.5 in physiological saline. To these cultures (5 mL), juglone was added at final concentrations 0, 100, and 200 μg/mL. These were then incubated at 37 °C, after which cells were separated from supernatants by centrifugation (8000× *g*, 10 min) followed by filtration of supernatants through 0.22 μm pore-size PVDF membranes. The concentration of potassium ions in the cell-free supernatant was then determined by atomic absorption spectrophotometry (ZEEnit700P, Jena, Germany). In addition, KCl solutions of different concentrations were used as the standard curve.

### 2.7. Membrane Potential Determinations

Changes in membrane potential were determined as previously described [25], with some modifications. Briefly, *L. monocytogenes* was cultured for 18 h in 30 mL of TSB at 37 °C, harvested by centrifugation (8000× *g*, 10 min), rinsed twice, and resuspended at a concentration of OD_600nm_ = 0.5 in physiological saline. These cultures (125 μL, OD_600nm_ = 0.5) were supplemented with juglone (125 μL) to final concentrations of 0, 50, 100, and 200 μg/mL. The cultures were incubated at 37 °C for 30 min in black 96-well microtiter plates. Subsequently, cultures (250 µL) were mixed with the fluorescent probe bis-(1,3-dibutylbarbi-turic acid) trimethine oxonol (DiBAC4(3); Molecular Probes, Solarbio, Beijing, China) at the final concentration of one µM and transferred into individual wells of a black 96-well plate, then incubated for five minutes at 37 °C in the dark. After five minutes, fluorescence intensities were measured at excitation wavelength 492 nm and emission wavelength 515 nm with a multimode reader (Spectra Max M 2, Molecular Devices, San Jose, CA, USA). Background fluorescence caused by medium supplemented with juglone was also measured and used to correct fluorescence intensities.

### 2.8. FEG-SEM Assay

Samples for FEG-SEM assays were prepared using a modified, previously described protocol [26,27]. Cultures (approximately 10^6^ CFU/mL) were supplemented with juglone at final concentrations of 0, 50, 100, and 200 μg/mL. Cells were incubated at 37 °C for 4 h, centrifuged (8000× *g*, 10 min, 4 °C), then rinsed twice with phosphate-buffered saline (PBS) (pH7.2), and lastly fixed for 12 h with 2.5% glutaraldehyde at 4 °C. After centrifugation (8000× *g*, 10 min), the cells were rinsed twice in PBS (pH 7.2) and dehydrated in ethanol series (30%, 50%, 70%, 80%, 90%, and 100%) for 10 min. Finally, the samples were placed at critical points to dry and subjected to gold spray treatment in vacuo. Cells were observed under FEG-SEM (Nano SEM-450; FEI, Hillsboro, OR, USA).

### 2.9. CLSM Assay

The damage to *L. monocytogenes* cell membranes were observed using CLSM as described by Kang et al. [28], with some modifications. *L. monocytogenes* cells were collected per Section 2.2 and treated with juglone (0, 50, 100, and 200 μg/mL) at 37 °C for 3 h. Then, the cells were washed twice with sterile PBS, pH7.3, collected by centrifugation at 8000× *g* for 5 min, and resuspended in 200 μL working staining fluid (5 μM Syto-9 and 2 μg/mL PI). This was then mixed thoroughly at 37 °C in the dark for 15 min. Finally, the bacterial droplets were placed on a glass slide, covered with a coverslip, and scrutinized under CLSM (FV1200; Olympus, Tokyo, Japan) at 1000× magnification.

### 2.10. FT-IR Spectroscopy

Samples for FT-IR were prepared, according to the method of He et al. [29], with slight modifications. *L. monocytogenes*, prepared per Section 2.2, were incubated with juglone (50, 100, and 200 μg/mL) at 37 °C for 8 h, while control cells were treated with 2% ethanol (*v*/*v*) under the same conditions. The cell pellets were obtained after centrifugation (8000× *g*, 10 min) and rinsed twice with sterile PBS (pH 7.2) buffer. Samples were freeze-dried in vacuo for two days. Each sample (1.0 mg) was then mixed with dried potassium bromide powder (100 mg) and ground into a uniform powder for tableting. FT-IR spectras were obtained using a Bruker vertex70 FT-IR spectrophotometer (Bruker, Karlsruhe, Germany) at wavenumbers 400 cm^−1^~4000 cm^−1^ with a resolution of 4 cm^−1^ for 32 scans. Finally, acquired spectras were smoothed, normalized, and baseline-corrected.

### 2.11. SDS-PAGE

The effects of juglone on intracellular soluble proteins of *L. monocytogenes* were analyzed using a modified SDS-PAGE method [30]. *L. monocytogenes* were cultivated in media supplemented with specific concentrations of juglone (MIC, 2 MIC, and 4 MIC) at 37 °C for 8 h. Cultures in juglone-free media supplemented with 2% ethanol (*v*/*v*) were used as control. Samples were centrifuged for 5 min (8000× *g*, 4 °C) after incubation. Supernatants were discarded, and cell pellets were then rinsed twice with PBS (pH 7.2). The lysate preparation (50 mM Tris (pH 7.4), 150 mM NaCl, 1% Trion X-100, 1% sodium deoxycholate, 0.1% SDS, and protease and phosphatase inhibitor cocktail for bacterial cell extracts) was added into the cell pellet and incubated for 50 min on ice. The supernatant was collected by centrifugation (8000× *g*, 10 min, and 4 °C), and its protein concentration was determined using a BCA Protein Assay Kit (Solarbio, Beijing, China). Then, 10 μL SDS-PAGE loading buffer was added to 30 μL of samples. These mixtures were boiled for 8 min, cooled on ice, and then ran in SDS-PAGE gels. After electrophoresis, gels were stained with Coomassie brilliant blue R-250 and subsequently decolorized to visualize protein bands.

### 2.12. Statistical Analysis

All experiments were performed in triplicates. Data are expressed as means ± standard deviation. One-way analysis of variance (ANOVA) and Tukey’s test multiple range tests (SPSS 23.0, SPSS Inc., Chicago, IL, USA) were used to evaluate the significant differences between means (*p* < 0.05).

## 3. Results and Discussion

### 3.1. Antimicrobial Effects of Juglone

The antibacterial activity of juglone was evaluated by inhibition zone and MIC. The diameter of the inhibition zone formed by juglone on the surface of *L. monocytogenes* was shown in Figure 1a. Low concentrations of juglone (250 μg/mL) inhibited the growth of *L. monocytogenes*, with inhibition zones of diameter 11.09 mm. This inhibitory effect increased with concentration, and the diameter of the inhibition zone was 20.27 mm at 4000 μg/mL of juglone. The inhibitory concentrations of juglone against *L. monocytogenes* are summarized in Figure 1b and Table 1. Tubes 5–8 were turbid, and tubes 1–4 were clear, indicating that *L. monocytogenes* in tubes 1–4 have no visible growth. Thus, the MIC of juglone against *L. monocytogenes* was 50 μg/mL. Relatedly, Li et al. showed that the MIC of shikonin—another naphthoquinone compound—against *L. monocytogenes* was also 50 μg/mL, suggesting that compounds with similar structures have similar biological activities [31]. Previous studies have shown that carvacrol, thymol, clove oil, phenyl lactic acid, and olive oil polyphenol extract had respective MICs of 250, 250, 500, 1250, and 1250 μg/mL against *L. monocytogenes* [32,33,34,35,36]. Based on these previous reports, it appears that juglone has stronger antimicrobial activity against *L. monocytogenes* than other currently studied plant extracts.

### 3.2. Growth Curves

The growths of *L. monocytogenes* were completely inhibited by juglone at concentrations ranging from 50 μg/mL to 200 μg/mL, as per growth-curve analysis (Figure 2). The lag phase of *L. monocytogenes* was prolonged by juglone at 6.25, 12.5, and 25 μg/mL. However, at concentrations of 6.25 μg/mL or below, juglone had no effect on the growth of *L. monocytogenes*. These results suggest that the inhibitory effects increased in a concentration-dependent manner. Growth-curve analysis not only showed the lag period was significantly prolonged at subinhibitory concentrations but also corroborated 50 μg/mL of juglone as its MIC against *L. monocytogenes*. Li et al. (2021) reported a similar effect of shikonin on the growth of *L. monocytogenes* [31], further supporting the hypothesis that compounds with similar structures have similar biological activities.

### 3.3. Leakage of Intracellular Potassium Ions

The permeability of the cell membrane could be identified by quantifying the amount of potassium ions release [29,37]. The amounts of extracellular potassium ions of *L. monocytogenes* with different treatments are displayed in Figure 3. Compared to the untreated-with-juglone group, the amount of potassium ions released was significantly elevated for the juglone-treatment group. Concretely, the potassium ions leakage in 100, 200, and 400 μg/mL juglone treatment increased by 218%, 230%, and 264%, respectively, compared to the untreated-with-juglone group. Furthermore, the amount of potassium ions treated with different concentrations of juglone have a sharp rise in 30 min, indicating that juglone has a rapid destruction of the permeability of the cell membrane. This sharp rise could be ascribed to either elevated membrane permeability or depolarization of cell membrane induced damage on potassium channels [26,38]. However, different antimicrobial agents have different effects on the release of potassium ions from bacterium. Fang et al. (2022) reported that phenyl lactic acid promoted the release of potassium ions from *Vibrio parahaemolyticus* within five minutes [39]. On the other hand, β-type oligomeric procyanidins from a lotus seedpod could promote the release of potassium ions from enterotoxigenic *E. coli* in a longer process of about 6 h [40]. These differences in duration might be due to the different modes of action of the various antibacterial agents.

### 3.4. Membrane Potential

In analyses of changes in membrane potential, *L. monocytogenes* that had been cultivated in media supplemented with 50~400 μg/mL of juglone had a noticeable decrease in fluorescence in comparison with the untreated-with-juglone group (Figure 4). Moreover, the higher the concentration of juglone is, the lower the resultant fluorescence. Membrane potential depolarization and hyperpolarization are hallmarks of membrane damage [27]. DiBAC4(3) is a lipophilic anionic fluorescent dye commonly used to detect cell membrane potential [41]. It enters depolarized cells and increases fluoresce when bound to intracellular proteins but leaves hyperpolarized cells with decreased fluorescence [19]. Thus, our results indicate that *L. monocytogenes* treated with juglone displayed membrane potential hyperpolarization. This phenomenon is mainly caused by change of pH and excitation of ion motion, especially the diffusion of potassium ions or potassium ions with several other ions [27]. Similar research reported that alkyl ferulate esters can hyperpolarize the cell membranes of *L. monocytogenes* [42]. Relatedly, citral caused cell membrane hyperpolarization in *Yersinia enterocolitica* and *Cronobacter sakazakii* [23,43].

### 3.5. FEG-SEM-Based Observations of Cell Morphology

The antibacterial mode of action of juglone was further understood through the effect of juglone on the morphological changes of *L. monocytogenes* by FEG-SEM. As shown in Figure 5, *L. monocytogenes* cells cultivated in media supplemented with juglone (50 μg/mL, Figure 5B) had continuous pits on the bacterial surface and were more wrinkled and deformed compared to the controls, which were plump, short-rod-shaped, and smooth-surfaced (Figure 5A). Those from media supplemented with juglone (100 μg/mL) had deeper cell pits and some ruptured cell membranes (Figure 5C). Those from media supplemented with juglone (200 μg/mL) had completely destroyed membranes, lacked their original rod-shaped morphology, leaked cell contents, and generally were fragmented cells (Figure 5D). This revealed concentration-dependent damage of the *L. monocytogenes* membrane by juglone. Moreover, these findings show that juglone ruptured the cell membrane in an irreversible manner, resulting in a bactericidal effect. Linalool and noni fruit extract had the same effect on *L. monocytogenes* [44,45]. However, alkyl ferulate esters achieves their bacteriostatic effect by making holes in the surface of the cell membrane of *L. monocytogenes* [42]. Citral treatments led to *Yersinia enterocolitica* cell membrane atrophy and sagging but no membrane rupture nor the formation of pores [43]. Therefore, different natural compounds have different modes of action on different pathogens. Altogether, cell membrane damage is one of the targets of juglone against *L. monocytogenes*.

### 3.6. CLSM-Based Observations of Cell Membrane Injury

Cell membrane damage by juglone was confirmed by CLSM integrated with the fluorescent probes SYTO 9 and PI. SYTO 9 has membrane permeability and penetrates cells freely, resulting in emission of green fluorescence. PI only penetrates damaged cell membranes, resulting in emission of red fluorescence. There was a transition from green to red fluorescence as a function of the concentration of juglone treatment (Figure 6). *L. monocytogenes* cells in the juglone-free group emitted distinct green fluorescence only, indicating that they had intact cell membranes. After treatment with juglone (50 μg/mL), the green-fluorescence intensity diminished and small amounts of red fluorescence were observed, indicating cytoplasmic membrane injury in a small proportion of cells. After treatment with juglone (100 μg/mL), the green-fluorescence intensity further decreased, while the red-fluorescence intensity was further increased, indicating a larger proportion of cells had lost their membrane integrity. After treatment with juglone (200 μg/mL), the treated *L. monocytogenes* cells only emitted red fluorescence. This demonstrated that juglone exerted the membrane damage in a concentration-dependent manner. Relatedly, juglone damaged the cell membranes of *Candida albicans* and *Pseudomonas syringae* in a concentration-dependent manner [12,15]. Altogether, these findings support the previous hypothesis that juglone damaged the cell membranes of *L. monocytogenes*.

### 3.7. FT-IR Analysis

FT-IR spectroscopy was widely used to explored the antimicrobial mode of action and monitor the structural changes in bacterial cell membranes under various forms of stress [46,47]. The FT-IR spectra of bacterial cells are generally distinguished into five regions: lipids (3000–2800 cm^−1^), protein/amides I and II (1700–1500 cm^−1^, phospholipids/DNA/RNA (1500–1185 cm^−1^), polysaccharides (1185–900 cm^−1^), and the fingerprint region (900–600 cm^−1^) [46]. The FT-IR analysis of juglone-treated *L. monocytogenes* at peaks of 2928 cm^−1^ (representing the CH_2_ antisymmetric stretching of lipids) had band intensities that increased with increasing juglone concentration, which indicated an increase in the saturated lipid concentration of the membrane (Figure 7). The enhanced peak intensity of another fatty-acid-associated band (CH_2_ bending) at around 1455 cm^−1^ further supported the results. Increases in saturated lipid concentration are often explained as an increase in rigidity and permeability of the bacterial cell membrane [29,48]. These increments occur when the cellular membrane is damaged, leading to cell destruction and death [49,50]. Similar results were observed for *S. aureus* that were subjected to an ultrasound coupled with thyme essential oil nanoemulsions [29]. With increasing concentrations of juglone, *L. monocytogenes* cells showed a corresponding increase in spectral intensities in regions 1653 cm^−1^ (dominated by amide I; C–O hydrogen-bonded stretching vibrations), reflecting the proteinaceous contents of the cells [51]. This might be attributed to the denaturation of proteins in the cell walls and cell membranes after treatment with juglone. Furthermore, juglone treatment caused changes in bands at 1397 cm^−1^ (attributed to C=O symmetric stretching of COO- group in amino acids, fatty acids), 1235 cm^−1^ (attributed to phosphate group (P=O) asymmetric stretching of the phosphodiester bond of nucleic acid), and 1072 cm^−1^ (attributed to symmetric stretching of P=O phospholipids in cell membranes) [47,52]. These changes demonstrated that juglone resulted in either *L. monocytogenes* cellular membrane damage or some conformational and compositional changes in the molecular makeup of the cell membrane.

### 3.8. SDS-PAGE Analysis

Protein is closely related to the physiological functions of cells. The protein bands of the marker and cells were shown in Figure 8. After cultivation in media supplemented with juglone (2MIC and 4MIC), bacterial protein bands of *L. monocytogenes* were fainter and even disappeared when compared to those of the control. This implied that juglone inhibited *L. monocytogenes* by affecting protein synthesis. This is similar to the effect of ε-Poly-lysine and lactic acid on *L. monocytogenes* [53,54]. Protein bands between 20–25 KDa in *L. monocytogenes* treated with juglone at the MIC, 2MIC, and 4MIC were increased more significantly than those of the control, as were those of 50–70 kDa and 25–40 kDa from cells treated with juglone at the MIC. Previous studies have found that bacteria can synthesize small proteins or molecular chaperones to strengthen resistance to external pressure and protect cells from protein denaturation when faced with external stress [55]. Thus, we hypothesize that *L. monocytogenes* synthesize some stress proteins to strengthen resistance to the effects of juglone.

## 4. Conclusions

In conclusion, the present research elucidated the antibacterial activity and mechanism of action of juglone against *L. monocytogenes*. The antibacterial mechanism of action of juglone against *L. monocytogenes* included increased cytomembrane permeability, cell membrane hyperpolarization, destroyed proteins or inhibited proteins synthesis, changes in membrane integrity, and, ultimately, complete disintegration of the cells. Overall, our findings show that juglone is a potential antibacterial agent, which is, thus, an alternative or supplemental strategy to mitigate the infections caused by *L.*
*monocytogenes*. Further research should focus on the effect of juglone on the antimicrobial signaling pathways of *L.*
*monocytogenes* by transcriptome and proteomics analyses.

## Figures and Tables

**Figure 1 foods-11-02558-f001:**
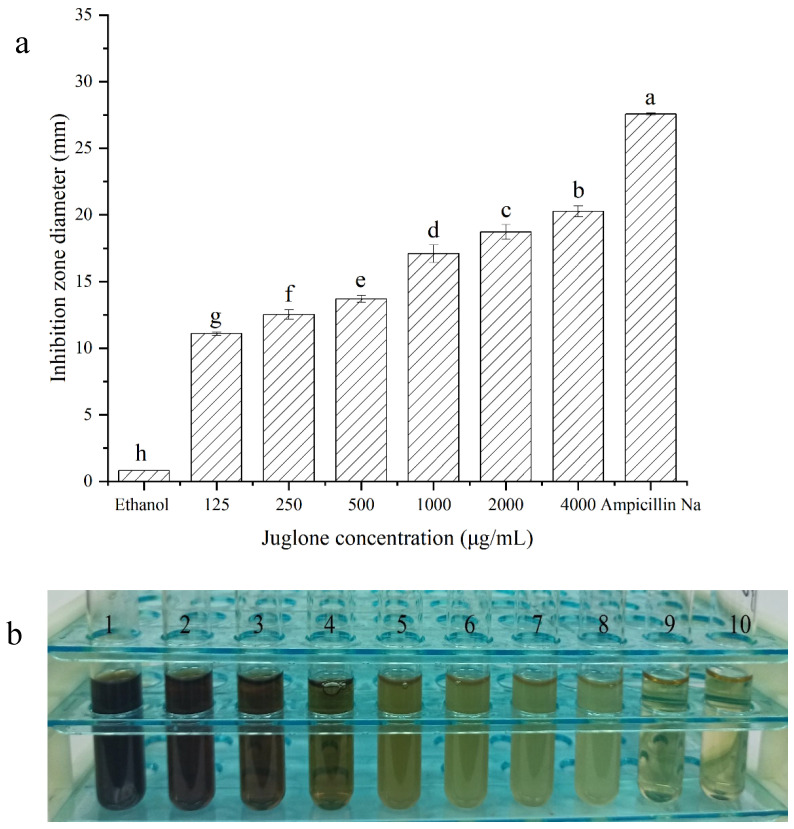
Antibacterial activity of juglone against *L. monocytogenes*. (**a**) Inhibition zones in *L. monocytogenes*. (**b**) Minimum inhibitory concentration of juglone against *L. monocytogenes*. Bars represent the standard deviation (*n* = 3). Different letters denote significant differences (*p* < 0.05). Note: 1–10: 400 μg/mL, 200 μg/mL, 100 μg/mL, 50 μg/mL, 25 μg/mL, 12.5 μg/mL, 6.25 μg/mL, 2% ethanol, TSB, ampicillin Na (50 μg/mL).

**Figure 2 foods-11-02558-f002:**
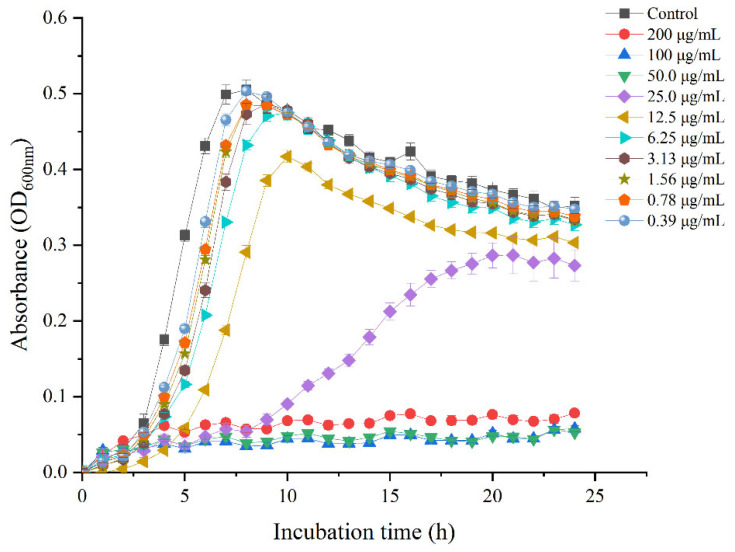
Growth of *L. monocytogenes* in TSB supplemented with various concentrations of juglone. Bars represent the standard deviation (*n* = 3).

**Figure 3 foods-11-02558-f003:**
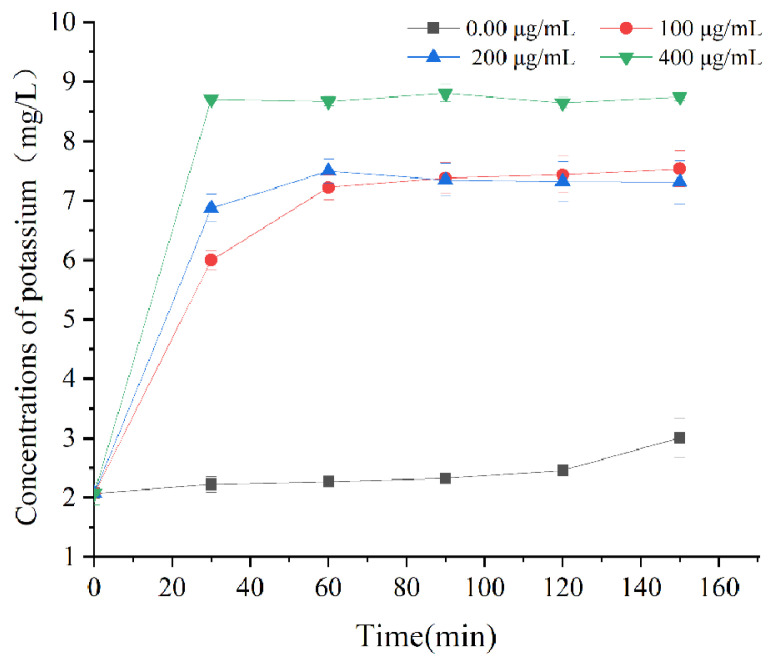
Effects of juglone on the release of potassium. Error bars indicate the standard deviation of three technical replicates (*n* = 3).

**Figure 4 foods-11-02558-f004:**
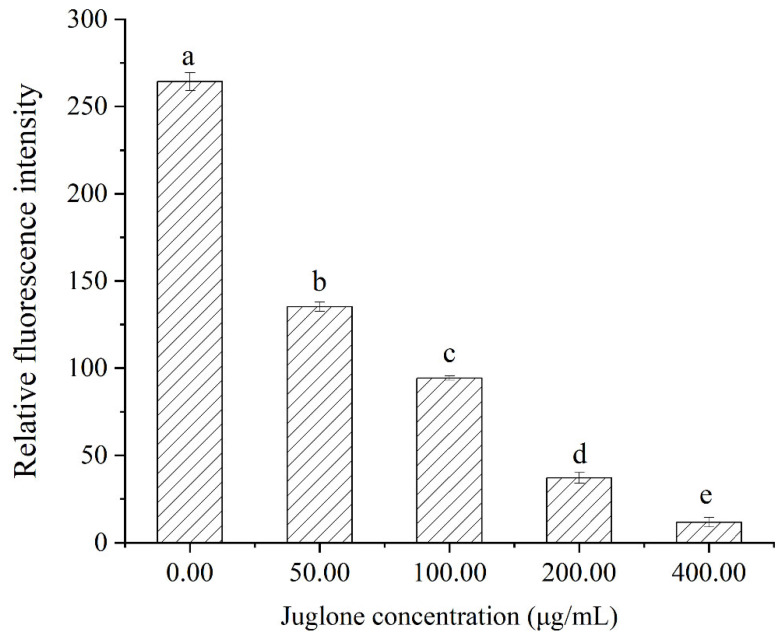
Effects of juglone on the membrane potentials of *L. monocytogenes*. Values represent the means of independent triplicates. Bars represent the standard deviation (*n* = 3). Different letters denote significant differences (*p* < 0.05).

**Figure 5 foods-11-02558-f005:**
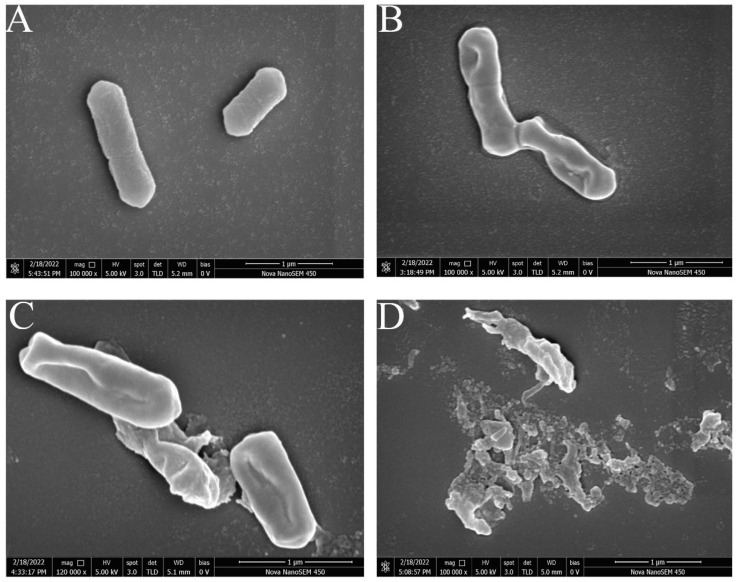
Scanning electron micrographs of *L. monocytogenes* untreated for 4 h (**A**) and treated with juglone at 50 μg/mL (**B**), 100 μg/mL (**C**), and 200 μg/mL (**D**) for 4 h. Note: bar scale is 1 µm, magnification 100,000×.

**Figure 6 foods-11-02558-f006:**
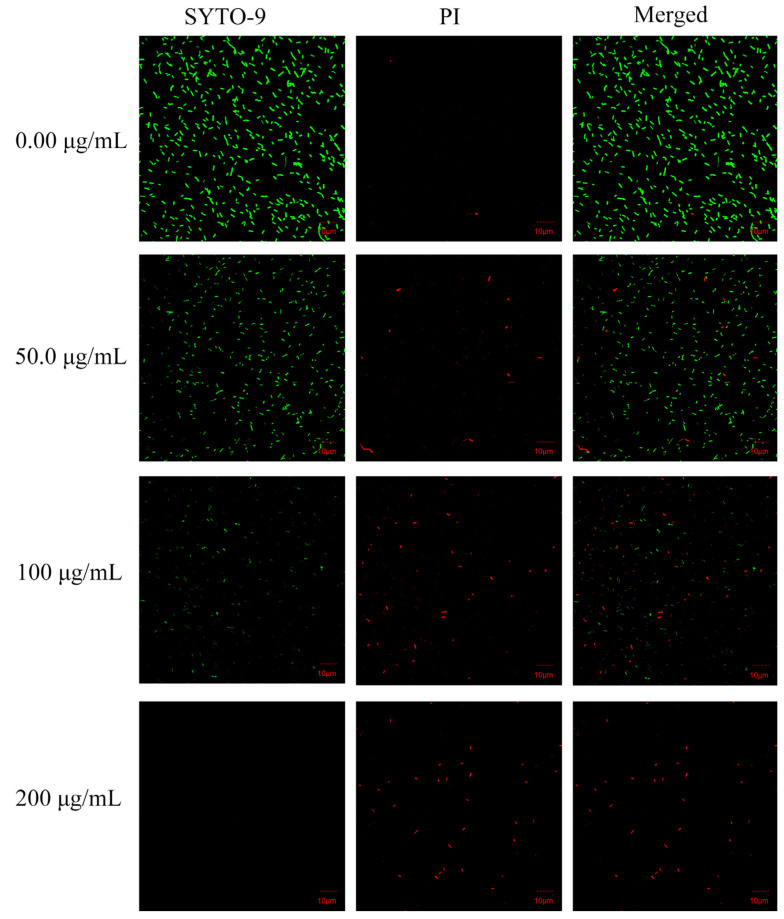
Confocal laser scanning microscopy of *L. monocytogenes* untreated with juglone and treated with juglone at 50 μg/mL, 100 μg/mL, and 200 μg/mL.

**Figure 7 foods-11-02558-f007:**
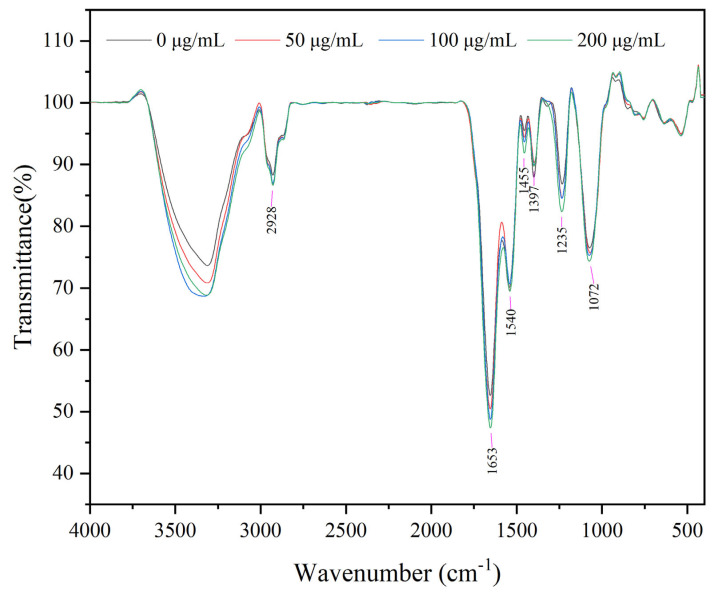
FT-IR images of *L. monocytogenes* treated with different concentrations of juglone.

**Figure 8 foods-11-02558-f008:**
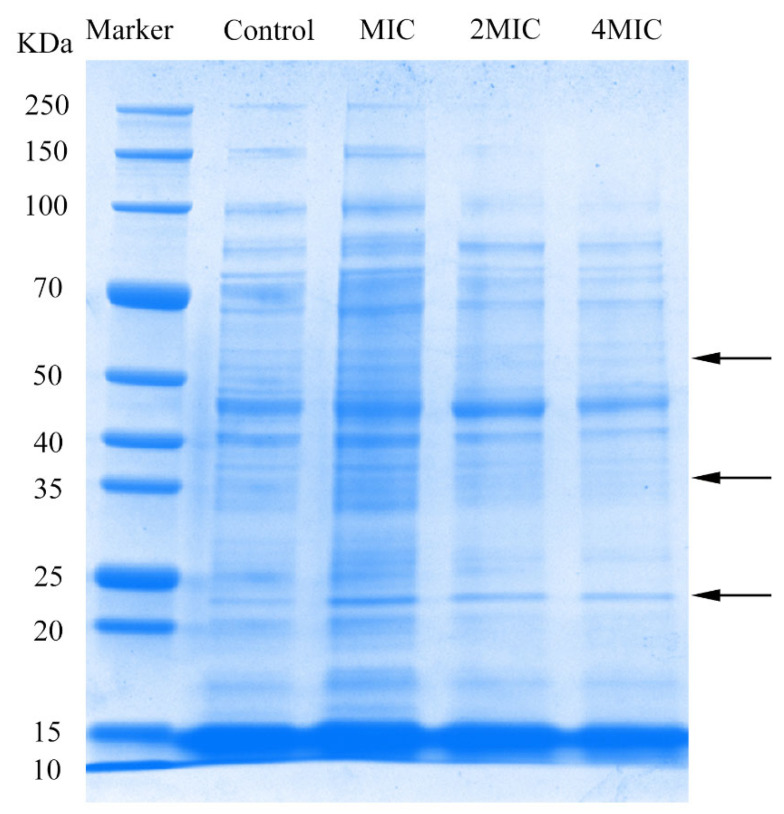
SDS-PAGE analysis of intracellular soluble proteins of *L. monocytogenes* not treated with juglone and treated with juglone at MIC, 2MIC, and 4MIC; the arrows are mainly to illustrate the changes of protein bands. The molecular weight range represented by the arrows corresponds to the Marker on the left.

**Table 1 foods-11-02558-t001:** Growth of *L. monocytogenes*.

Bacteria	Concentration of Juglone (µg/mL)							Blank Control	Negative Control	Positive Control
	6.25	12.5	25	50	100	200	400			
*L. monocytogenes*	+	+	+	-	-	-	-	-	+	-

Note: “+”: indicates observed growth of bacteria, “-” indicates no visible growth of bacteria.

## Data Availability

Data is contained within the article and is available at request from the corresponding author.

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
