# Peer review of "Juglone Inhibits Listeria monocytogenes ATCC 19115 by Targeting Cell Membrane and Protein"

_foods, 2022, doi:10.3390/foods11172558_

Round 1
Reviewer 1 Report
Dear Authors,
The manuscript (ms id: foods-1859678) by Y. Cai et al entitled “Juglone inhibits Listeria monocytogenes by targeting cell membrane and protein” is an interesting study on the antimicrobial activity of Juglone which is yielded by the walnut green husk on L. monocytogenes causing foodborne disease. To elucidate its antibacterial mechanism, the authors determined the resultant morphology, membrane permeability, membrane integrity, and cell proteins by the action of juglone against L. monocytogenes. They examined classical antibacterial tests, the inhibition zones, the minimum inhibitory concentrations, and the growth curve analyses of L. monocytogenes to assess its antimicrobial properties. In addition, they also examined the membrane permeability and integrity by assessing the leakage of potassium ions, membrane potential changes, and several observations of CLSM, FEG-SEM, and FT-IR. The experiments are almost properly conducted and presented, and their methods are clear. The data results are represented and interested in readers of the journal. In general, the manuscript provides a valuable insight to control foodborne L. monocytogenes infections.
Major comments:
(1) For SDS-PAGE analysis of intracellular soluble proteins, I think those results and discussion are weak evidence. Why did you compare only soluble proteins? Cell permeability and morphology observations indicate the cell membrane damage by adding Juglone. There is no mention of cell membrane proteins and insoluble proteins. Changes in protein bands in the Juglone treated cells are not identified, for example, partial N-terminal amino acid sequences or immunohistochemical properties. In addition, the molecular weights of the protein bands in the text are not represented in correspond to those in Fig. 8. It should be indicated using arrows.
(2) About the methodology of statistical analysis in the Materials and methods, the post hoc -test used in this study is Duncan’s multiple range tests to evaluate the significant differences between means. It has been pointed out that Duncan’s multiple range tests (DMRT) often give the risk of type-I error. Thus, it has been improved as a new DMRT methodology. Which is used in this study?
(3) Regarding Fig. 1a, I think the pictures presented in Fig. 1a are not necessary because their inhibition zone could not see.
Minor comments:
(1) In section 2.1. in Materials and Methods and Fig. 1a, the juglone concentration unit represents mg/mL but other experiment methods use mg/mL. Unless there is a specific reason, it should be changed to the same units.
(2) Regarding Table 1, the usage of uppercase and lowercase letters should be the same.
(3) In section 3.2. Growth curves in the Results and Discussion. “These results suggest that the inhibitory effects decreased in a concentration-dependent manner.” I think Fig. 2 shows the Juglone concentration-dependent increasing inhibitory effect. Thus, I think the above statement is wrong.
(4) Fig. 2. The Y-axis title should be reconsidered. Optical density at 600 nm does not present real cell numbers.
(5) Please double-check the list of references. “Listeria monocytogenes” should be italic letters. In several references, the doi number is broken in the middle of a number. Therefore, the total number of reference articles in the reference list does not match the number of references in the text of the manuscript.
Reviewer 2 Report
Manuscript Foods 1859678
This manuscript is about how the juglone affect the membrane disruption of Listeria monocytogenes causing its death. Is very interesting how they demonstrated the action mechanism of this compound.
In general, the manuscript is clear to understand and I have some suggestions for the edition of it.
General comments
Figure 2. In this figure the authors should change the name of the Y axis, it should be "Absorbance (OD 600nm)" instead of Numbers of bacteria. OD were measured, bacteria were not counted.
The manuscript has not the line’s numbers. In this paragraph “Concretely, the potassium ions leakage in 100, 200 and 400 μg/mL juglone treatment increased by 218%, 230%, and 264% respectively…” please change the percentages for the differences in mg/L units.
In Figure 5, the letters A, B, C and D are black, they need contrast, please change to a white color.
Reviewer 3 Report
The manuscript entitled "Juglone inhibits Listeria monocytogenes by targeting cell membrane and protein" is very interesting, well written and the experimental design is appropriate;
The authors tested only L. monocytogenes ATCC 19,115; considering that L. monocytogenes strains show considerable variability towards antimicrobial compounds, the authors must indicate the tested strain in the title “Juglone inhibits Listeria monocytogenes ATCC 19,115 by targeting cell membrane and protein “
Abstract
(4-45℃) 4/45 °C
1. Introduction
Finally, we explored the effects of juglone on the pathogenesis of L. monocytogenes……
I think it is more correct to speak of pathogenicity, virulence, not pathogenesis of L. monocytogenes
2.7. ….Then, the cells were washed twice with sterile PBS (pH7.3), pH 7.3
